# 3D Bioprinting for Vascularized Tissue-Engineered Bone Fabrication

**DOI:** 10.3390/ma13102278

**Published:** 2020-05-15

**Authors:** Fei Xing, Zhou Xiang, Pol Maria Rommens, Ulrike Ritz

**Affiliations:** 1Department of Orthopaedics and Traumatology, Biomatics Group, University Medical Center of the Johannes Gutenberg University, Mainz 55131, Germany; 2017324025214@stu.scu.edu.cn (F.X.); pol.rommens@unimedizin-mainz.de (P.M.R.); 2Department of Orthopaedics, West China Hospital, Sichuan University, No. 37 Guoxue Lane, Chengdu 610041, China; xiangzhou@scu.edu.cn; 3Trauma Medical Center of West China Hospital, Sichuan University, No. 37 Guoxue Lane, Chengdu 610041, China

**Keywords:** 3D bioprinting, bone regeneration, vascularization, tissue engineering, bioinks

## Abstract

Vascularization in bone tissues is essential for the distribution of nutrients and oxygen, as well as the removal of waste products. Fabrication of tissue-engineered bone constructs with functional vascular networks has great potential for biomimicking nature bone tissue in vitro and enhancing bone regeneration in vivo. Over the past decades, many approaches have been applied to fabricate biomimetic vascularized tissue-engineered bone constructs. However, traditional tissue-engineered methods based on seeding cells into scaffolds are unable to control the spatial architecture and the encapsulated cell distribution precisely, which posed a significant challenge in constructing complex vascularized bone tissues with precise biomimetic properties. In recent years, as a pioneering technology, three-dimensional (3D) bioprinting technology has been applied to fabricate multiscale, biomimetic, multi-cellular tissues with a highly complex tissue microenvironment through layer-by-layer printing. This review discussed the application of 3D bioprinting technology in the vascularized tissue-engineered bone fabrication, where the current status and unique challenges were critically reviewed. Furthermore, the mechanisms of vascular formation, the process of 3D bioprinting, and the current development of bioink properties were also discussed.

## 1. Introduction

Treatment of large bone defects resulting from cancer, trauma, infection, congenital malformation, or surgical resection is a challenge for clinical doctors [1]. Currently, the autologous bone grafts are the gold treatment standard for large bone defects [2]. However, the amount of autologous bone grafts is limited, and complications at the harvesting site, such as pain, infection, or bleeding, could result in additional donor-site morbidity [3]. Bone grafts fabricated by the tissue-engineered methods are rapidly becoming promising alternatives [4]. However, oxygen and metabolic needs are not met when the thickness of the tissue-engineered bone constructs exceeds 150–200 µm, resulting in lacking tissue integration with host tissue and core ischemia of tissue-engineered bone grafts [5,6]. Furthermore, insufficient vascularization of tissue-engineered bone often results in poor bone regeneration [7]. Therefore, during the fabrication of tissue-engineered bone constructs, it is vital to construct functional vascular networks, which could supply the exchange of nutrition, oxygen, and waste products between bone grafts and host [8].

Bone, being a dynamic tissue, is not only a complex heterogeneous tissue with intricate hierarchical architecture but also a highly ordered and vascularized tissue with vascular networks, which is connected to the blood system by transverse channels [9,10,11]. Traditional tissue-engineered methods based on seeding cells into the scaffold could not precisely control the inner structure, cell distribution, and exocellular microenvironment to meet the biomechanical functions and metabolic requirement of bone tissue [12]. In addition, traditional tissue engineering methods cannot fabricate the biomimetic tissue-engineered constructs with realistic cell microenvironment, which leads to over-simplified tissue-engineered constructs [13,14]. Therefore, the limitations of traditional tissue engineering technologies to recreate similarities and complexes from native bone tissues restrict their further applications [15]. Nanomaterials with bone-mimicking characteristics can construct proper cell microenvironments to enhance bone regeneration [16]. However, there are several challenges of nanomaterials, such as failure to temporospatial administration of growth factors and cells, as well as lack of integrative networks of new bone tissues and blood vessels [17].

Three-dimension (3D) printing technology, initially introduced in 1986, has been widely used to fabricate objects with complex geometries and architecture [18]. Recently, 3D bioprinting technology has emerged as a promising alternative to fabricate 3D functional tissue constructs with geometrically defined structures, which are designed to replace or regenerate the damaged tissues or organs, such as liver, bone, skin, liver, cartilage, nerve, and heart [19]. Three-dimensional (3D) bioprinting is the use of a combination of 3D printing, tissue engineering, developmental biology, and regenerative medicine to construct biomimetic tissues. Particularly, compared to conventional scaffold-based approaches, 3D bioprinting technology could precisely control complex 3D architecture, multiple compositions, and spatial distributions [13]. Both 3D printing and 3D bioprinting could utilize the layer-by-layer manner to fabricate 3D anatomically shaped constructs from a computer-aided design (CAD) model. However, 3D bioprinting technologies involve the utilization of cell-laden bioinks and other bioactive molecules to fabricate biomimetic tissue constructs during the printing process, while 3D printing technologies do not involve the utilization of cells or other bioactive molecules [13,20]. In addition, 3D bioprinting technology can guide tissue formation for patient-specific therapy by precise spatiotemporal control on the distribution of cells, growth factors, small molecules, drugs, miRNA, and other bioactive substances [21,22]. Therefore, we conducted this review using EMBASE, PubMed, Medline, and Web of Science for studies on the application of 3D bioprinting technology in the vascularized tissue-engineered bone fabrication. This review discussed the current status and unique challenges. Furthermore, the mechanisms of vascular formation, the process of 3D bioprinting, and the current development of bioink properties were also discussed.

## 2. Mechanisms of Vascular Formation

New blood vessels are formed by two main fundamental processes, angiogenesis and vasculogenesis [23]. Vasculogenesis is known as differentiation of endothelial progenitor cells (EPCs) or angioblastic progenitor cells into endothelial cells (ECs) and the formation of a primitive vascular network [24]. During the process of early embryo development, primitive capillary networks form through recruitment and differentiation of angioblastic progenitor cells [25]. In adults, vasculogenesis was often observed in the repair process of various damaged tissues and pathological states such as atherosclerosis, ischemia, and tumor [26]. Angiogenesis is known as the growth process of new capillaries from pre-existing blood vessels. Furthermore, the process of angiogenesis is tightly regulated by related biological factors [27]. When the initial vascular network forms into more complex vascular networks, vasculogenesis is followed by angiogenesis. Angiogenesis plays a vital role during the process of tissue regeneration. Angiogenesis disorders are implicated in the pathogenesis of a variety of diseases, including vascular retinopathy, rheumatoid arthritis, and tumor [28]. After the process of angiogenesis, the vascular network expands and bridges with other capillary networks, which is also called splitting angiogenesis. In the process of vessel maturation, smooth muscle cells and differentiated pericytes stabilize vascular structures and suppress the growth of ECs [26]. The process of vasculogenesis and angiogenesis is shown in Figure 1a.

Bones are highly vascularized and receive around 10%–15% of resting cardiac output [31]. The microvasculature of bone tissue is consisted of three types of vessels, namely capillaries, arterioles, and venules [32]. The schematic illustration of complex bone tissue with vascular structure is shown in Figure 1b. Vasculature networks play an important role in the process of bone tissue formation. As a coordinated process, bone tissue regeneration involve the connection between bone cells and blood vessels [33]. The two general ways of bone tissue formation, intramembranous and endochondral ossification, are also regulated by the bone vasculature [34]. During the process of endochondral ossification, mesenchymal progenitor cells aggregate into dense clusters and differentiate into chondrocytes. The nonproliferative chondrocytes in the cartilage template secrete proangiogenic factors, which stimulate blood vessels to invade and, along with osteoclasts and osteoprogenitors, to form the primary ossification center [35]. As the vasculature expands, the cartilage is replaced with bone tissue, resulting in the growth of longitudinal bone [36]. Unlike endochondral ossification, intramembranous ossification is the development process of bone tissue from fibrous membranes. Microcapillary network growth extends into the mesenchymal region of periosteum, resulting in the differentiation of mesenchymal cells into osteoprogenitors and osteoblasts [37]. Currently, two strategies are used to construct the vascular network in engineered bone, including creating a major vessel with ECs and the formation of microcapillaries through self-assembly and biological processes [38,39]. Ideally, intramembranous and endochondral ossification should be combined to fabricate tissue-engineered bone with multiscale vasculatures.

## 3. 3D Bioprinting

### 3.1. The Procedure of 3D Bioprinting

As an emerging multidisciplinary subject, 3D bioprinting consists of 3D printing technology, tissue engineering, developmental biology, regenerative medicine [32]. More specifically, 3D bioprinting technology is an additive manufacturing process of tissue-like structures by utilization of 3D printing-like techniques to combine biomaterials, cells, and growth factors [40]. The 3D bioprinting technology utilizes the layer-by-layer manner to deposit biomaterials, also known as bioinks, to fabricate tissue-like structures in tissue engineering fields [41]. Three-dimensional (3D) bioprinting technology can offer precise control on complex 3D architecture, spatial distributions, multiple compositions [42]. The first step of 3D bioprinting is to image the tomographic properties and functions of the target tissue by magnetic resonance imaging (MRI), computed tomography scanning (CT scan), and ultrasound imaging techniques (UI) [43]. The second step is to design and reconstruct precisely 3D functional tissues by a computer-aided design (CAD) model [44]. The next step is the tissue designs, which includes material selection and cell selection. Biomaterials, also known as bioinks, mimic the structure, shape, architecture, and function properties of the extracellular matrix of target tissues [45]. Additionally, bioinks play an essential role in supporting the adhesion, proliferation, and function of encapsulated cells [46]. Following cells suspended in bioinks, the cell-laden bioinks are then utilized to fabricate 3D biomimetic tissue constructs with geometrically defined structures by a bioprinter [47]. The last step of bioprinting procedure is the maturation phase of the engineered tissues or organs. The perfusion bioreactors containing nutrient transport and physiological stimuli mimic the environment and stimuli of natural tissue and promote the maturation of the engineered tissues [18]. During this maturation phase, the printed structures aggregate to form bigger continuous structures and place them precisely in the appropriate position [13]. Additionally, a 3D bioprinting tissue can be printed in situ, in which case the human body acts as the bioreactor. Figure 2 shows the general step-wise procedure for bioprinting 3D tissues.

### 3.2. 3D Bioprinting Methods in Fabrication of Vascular Networks

Briefly, according to their working mechanism, the typical procedures of 3D bioprinting technologies can be broadly classified into laser-assisted bioprinting, inkjet bioprinting, and extrusion-based bioprinting [48]. The simplified procedures of different kinds of 3D bioprinting are shown in Figure 3. Each of these approaches has been summarized in Table 1.

#### 3.2.1. Inkjet Bioprinting

Inkjet bioprinting is a rapid and large-scale fabrication technique that has been adapted from the inkjet printing technology to print living cells by desktop inkjet printers [76] (Figure 3a). Currently, many studies utilize inkjet bioprinting to construct various engineered tissues with different kinds of living cells [77,78,79]. In addition, inkjet bioprinting can rapidly fabricate tissue-like structures with intricate hierarchical architectures by the utilization of the controlled dropwise deposition of cell-laden bioinks [79]. As a non-contact printing technique, inkjet bioprinting could precisely deposit droplets of cell-laden bioinks in the z-axis onto the surface of culture dish or hydrogel substrates [80]. According to the droplet actuation mechanism, the inkjet bioprinting could be further classified into thermal inkjet printing and piezoelectric inkjet printing [81]. Thermal inkjet printing dispensed the bioink droplet by a thermal actuator with a voltage pulse to locally heat the bioink. In the process of piezoelectric inkjet bioprinting, the piezoelectric actuator can be activated by variations in the electric impulse and amplitude and used to control the stress to which the bioink is exposed [54]. As a nozzle-based technique, inkjet bioprinting had advantaged of high printing speed, affordability, high resolution [5]. However, the inkjet bioprinting is limited by the fact that only low-viscosity bioinks can be used for bioprinting; an additional crosslinking step in the process of inkjet bioprinting is needed to improve the structural stability of 3D tissue constructs [82].

Currently, several studies utilize inkjet bioprinting to create customized tissue-engineered with vascular networks. Xu et al. successfully used the 3D inkjet bioprinting system to fabricate 3D complex tissue-engineered constructs with fibroblast-based tubes [83]. Lee et al. fabricated a perfused vascular channel within thick collagen scaffold by inkjet bioprinting. Fully covered by ECs, the functional vascular channel can not only support the viability of tissue up to 5mm in the distance under the physiological flow condition but also presents a barrier for both plasma protein and dextran molecule [84]. Besides, Lee et al. connected the multi-scale capillary network to the large perfused vascular channels through a natural maturation process [85]. In another study, thermal inkjet bioprinting was used to fabricated constructs with microvasculature by depositing bioink consisting of human microvascular endothelial cells and fibrin [5].

#### 3.2.2. Laser-Assisted Bioprinting

Laser-assisted bioprinting precisely deposit cell-laden bioinks in a 3D spatial arrangement by an energy source of laser radiation, which is highly monochromatic, focused, and coherent [86]. The laser-bioprinting setup mainly consists of a laser source, a receiving substrate, and a ribbon coated with cell-laden bioinks [87] (Figure 3b). During the procedure of laser-assisted bioprinting, the ribbon is illuminated by a focused laser beam. As a consequence, cell-laden bioinks evaporate and reach onto the surface of receiving substrate, which supports the adhesion and proliferation of cells [88]. The lasers used in the laser-assisted bioprinting are mainly nanosecond lasers with UV [89]. The procedure of laser-assisted bioprinting is contactless, resulting in high post-printing cell viabilities. Additionally, laser-assisted bioprinting can not only print various living cells but also peptides and DNA [90]. Currently, few studies apply laser-assisted bioprinting technology to fabricate 3D vascularized tissue-engineered constructs. Wu et al. utilized laser-assisted bioprinting technology to construct a branch/stem structure of umbilical vein smooth muscle cells and umbilical vein endothelial cells (HUVECs) [91]. However, the branch/stem structure fabricated by laser-assisted bioprinting is a very simplified structure and could not mimic the structure of human vascular networks. Gruene et al. [92] utilized human adipose-derived stem cells (hASCs) to fabricate the 3D tissue grafts by the laser-assisted bioprinting technology. In addition, they also demonstrated that the procedure of laser-assisted bioprinting did not affected the proliferation ability and differentiation behavior of the hASCs. In another study, ECs were bioprinted by laser-assisted bioprinting technology onto a collagen hydrogel scaffold previously seeded with mesenchymal stem cells (MSCs) to fabricate a microvascular network [93]. Kérourédan et al. utilized laser-assisted bioprinting to bioprint ECs in situ into mouse calvarial bone defects prefilled with collagen scaffold containing MSCs and vascular endothelial growth factor (VEGF) [94]. The results demonstrated that in vivo laser-assisted bioprinting is a valuable approach to introduce in situ prevascularization with a defined architecture and enhance bone tissue regeneration.

#### 3.2.3. Extrusion-Based Bioprinting

Among these bioprinting technologies, extrusion-based bioprinting is a widely used approach of the material-dispensing technique used for bioprinting [95]. Extrusion-based bioprinting utilizes extrusion of the bioinks through a microscale nozzle to fabricate tissue-engineered constructs onto a stationary substrate [96] (Figure 3c). The extrusion is controlled using pneumatic pressure or mechanical compressions. After the layer-by-layer application, extrusion-based bioprinting fabricates 3D patterns and constructs [97]. Advantages of extrusion-based bioprinting are the direct incorporation of cells, processing at room temperature, and homogenous distribution of cells. Additionally, extrusion-based bioprinting has been utilized in the printing of cells and tissues with defined retention of activity [98]. Furthermore, tissue spheroids can also be loaded in pipettes of extrusion-based bioprinting to fabricate artificial tissues and organs [99]. In contrast to laser-assisted bioprinting limited by scalability and inkjet bioprinting limited by low viscosity, extrusion-based bioprinting can fabricate large, scalable tissue-engineered constructs with a wide range of viscosities. Due to its ease of control, cost-effectiveness, and the availability of shear-thinning bioinks, extrusion-based bioprinting is widely used [100]. However, the extrusion-based bioprinting is limited by the fact that the printing speed is relatively low [101].

Extrusion-based bioprinting technology has be proven a great promise in fabricating vascularized constructs with multiple cell types by the incorporation of several extrusion nozzles [102]. Tan et al. utilized the multi-nozzle extrusion-based technique to print vertically alginate-based tubular structures with varying viscosity [103]. The results demonstrated the feasibility of extrusion-based bioprinting to fabricate large diameter vascularized constructs. Extrusion-based bioprinting combined with microfluidic techniques can construct sophisticated 3D architectures in complex, heterogeneous constructs. Zhang et al. utilized chitosan and alginate hydrogels as bioinks to fabricate a printable vessel-like microfluidic channels by extrusion-based bioprinting [104]. Dolati et al. printed an alginate vascular conduit by a coaxial bioprinting process. In addition, the mechanical properties of the vascular conduit were enhanced by multi-walled carbon nanotubes [105]. Researchers also utilized this combined bioprinting technique to integrate micro-engineered vasculature and cellular layers within the deposited extracellular matrix of target tissues [106]. Gao et al. also used a coaxial nozzle to print vessel-like hollow filaments [98]. Colosi et al. utilized this combined bioprinting technique to fabricate the tissue-engineered constructs with human umbilical vein endothelial cell–lined vasculature by depositing different bioinks using a blend of alginate and gelatin methacrylate (GelMA) [107].

### 3.3. Bioinks

In the field of bioprinting, bioinks is another important part [100]. The bioink is either a solution or a hydrogel of biomaterials encapsulating the desired cells and used for fabricating tissue-engineered constructs. The bioinks provide stable 3D architecture to affect the development and maturation of tissue [54] and mimic the tissue niche in situ. Therefore, bioink design plays a crucial role in the process of 3D bioprinting. Currently, various natural and synthetic biomaterials with different physical and chemical properties have been formulated and utilized as bioinks [108]. The 3D architecture of bioinks affects the phenotype of encapsulated cells, resulting in the activation of various cellular signaling pathways and the expression of various related genes [109]. Additionally, 3D bioprinting technology can combine two or more bioinks of distinct materials to construct hybrid scaffolds. Therefore, the development of bioinks still needs significant research to achieve better cell regulation function. An ideal bioink should possess a range of properties as follow: (i) good printability withstanding forces applied during the printing process; (ii) biocompatibility mimicking the natural microenvironment of the target tissues; (iii) structural stability and biodegradation; (iv) mechanical properties; (v) suitability for chemical modifications to meet tissue-specific needs. Furthermore, standardized bioink formulations are also required in bioprinting to be applied in different kinds of tissue. 

According to the different source, bioinks used in 3D bioprinting can be basically divided into two types of hydrogels: nature-derived and synthetic bioinks [50,110]. Bioinks from various sources exhibit different biological characteristics during the process of 3D bioprinting [111,112,113]. Nature-derived bioinks have been widely used to fabricate the tissue-engineering constructs, which could support the attachment and proliferation of bioprinted cells [75]. Most of the nature-derived bioinks are derived from the natural extracellular matrix of different tissue. Compared to synthetic bioinks, nature-derived bioinks more closely resemble the native tissue and provide a better cell microenvironment. In addition, nature-derived bioinks can provide tissue-specific nutrients for cells. Nature-derived bioinks commonly used in the 3D bioprinting include silk, chitosan, decellularized extracellular matrix, hyaluronic acid, fibrin, collagen, gelatin, hydroxyapatite, and alginate [50]. In contrast to nature-derived bioinks, synthetic bioinks are fabricated by the process of chemical synthesis. They are more controllable than nature-derived bioinks in terms of chemical and mechanical properties, such as alignment, porosity, tensile strength, and elastic modulus. Synthetic bioinks commonly used in 3D bioprinting include polyethylene glycol (PEG), polycaprolactone (PCL), pluronic acid [114,115]. The characteristics of various bioinks commonly used in the bioprinting are shown in Table 2.

Among these various bioinks, gelatin methacrylate (GelMA) is the most widely used bioink to fabricate the vascular network in engineered constructs [125,126]. Compared to ionically crosslinked alginate, GelMA can form a chemically stable hydrogel scaffolds when exposed to ultraviolet [127]. GelMA is a photopolymerizable hydrogel and contains many natural cell binding-motifs which promote cell adhesion and cell migration within the GelMA matrix [128]. The hydration and biomechanical properties of GelMA can be regulated by changing the gel concentration and methacrylation degree. In addition, osteogenic cells could be encapsulated into GelMA with microchannels lined with ECs to fabricate vascularized tissue-engineered bone tissues [129,130]. GelMA can also be combined with other biomaterials to form hybrid bioinks for the fabrication of vascularization. Jia et al. fabricated perfused vascular structures by a 3D bioprinting technique based on cell-laden bioinks consisting of sodium alginate and GelMA [63]. The blended bioink was crosslinked by calcium ions [63].

### 3.4. Cells Used in Bioprinting

The choice of various cell types is another key element for 3D bioprinting. The cells and biomaterials interact with each other. Additionally, to mimic the function of target tissue on a macro scale, the cells used in bioprinting must have the ability to proliferate and mimic the physiological state of cells in vivo and in vitro [131]. Currently, various types of mammalian cells, such as osteogenic, as well as angiogenic cells, have been successfully used to fabricate vascularized tissue-engineered constructs [44]. To date, among many various osteogenic cells, MSCs are the most likely used cells in 3D bioprinting for the fabrication of tissue-engineered bone, owing to their differentiation potential and self-renewing capability [19,132,133]. MSCs are widely found in cord blood, umbilical cord, adipose tissue, and bone marrow [134]. Induced pluripotent stem cells (iPSCs) were first found in 2007 and were generated directly from a somatic cell [135]. IPSCs have the advantages of expandability, easy accessibility, and ability to differentiate into other cell types [136]. Moreover, MSCs were recently derived from iPSCs, which could overcome the inadequate source of autologous MSCs and cell aging [137]. Moreover, MSCs are highly sensitive to the microenvironment of bioinks, including chemical, physical, and biological cues. Therefore, the function of encapsulated stem cells can be regulated by changing the environment of bioinks [138]. Phillippi et al. engineered stem cell microenvironments by 3D bioprinting technology. Through this approach, the researchers engineered cell fate toward the osteogenic lineage [71]. During the procedure of bioprinting, cells could be arranged as individually or dispersed or encapsulated in the hydrogel precursor. Additionally, cells could be utilized to construct cell aggregates or microcarriers into bioinks [139].

#### Cell Viability

Currently, there are many kinds of cells used in the bioprinting process to fabricate tissue-engineered bone constructs, such as MSCs and ECs [21,125]. There is still a challenge in maintaining the viability of the cells encapsulated inside the bioinks The post-printing cell viability in bioinks of laser-assisted bioprinting, inkjet bioprinting, and extrusion-based bioprinting is shown in Table 2. Currently, most studies found the 3D bioprinted cells had high post-printing cell viability after a few hours or days [13,95]. However, there are few long-term studies focusing on cell viability after bioprinting. In addition, print pressure and print speed directly affect cell viability [22]. Shear and thermal stress during the bioprinting process also affect cell viability [86]. In addition, if used too frequently, piezoelectric technology could damage cell membrane, resulting in cell death [76]. Although the bioinks can protect the cells from being damaged by the potentially high shear stress during the bioprinting process, different kinds of bioinks exhibit differences in maintenance of the viability of cells. In addition, some kinds monomers and photo-initiators, used in the process of crosslinking, can also affects cell viability.

### 3.5. Multi-Materials Bioprinting

Bioinks, used in the process of bioprinting, are hydrophilic and high-molecular weight polymers with high water content, which can be cross-linked to form a 3D bioprinted construct [140]. In addition, most kinds of bioinks used in bioprinting are hydrogels, which have a certain drawback of weak mechanical property [141]. Multi-materials bioprinting has been used to improve the biomechanical property and structural integrity of 3D bioprinted constructs [142]. In addition, multi-materials bioprinting can be utilized to fabricate biomimetic and heterogeneous constructs, such as vascularized tissues [142]. The ability to deposit multi-materials materials in 3D bioprinting is consistent with support strategies found in the process of 3D printing. During the bioprinting process, support materials are utilized to enhance the structure integrity and mechanical property of 3D bioprinted constructs. Currently, the approaches of multi-materials bioprinting mainly include multi-head systems, core-shell needle systems, stereolithography, and multi-material microfluidic bioprinting [142,143]. However, how to improve printing resolution and integrate different kinds of materials still need to be solved in the application of multi-materials bioprinting.

## 4. 3D Bioprinting in the Fabrication of Vascularized Tissue-Engineered Bone

The vasculature in various tissues transports the required oxygen and nutrients and removes waste products. Vascularization plays an essential role in successful engineering of tissue constructs. Constructing vascular networks within 3D tissue-engineered bone constructs is a critical challenge in maintaining the viability of bioprinted cells. Currently, many approaches have been utilized to improve the growth of vascular networks within 3D tissue-engineered bone constructs. Currently, many researchers incorporate growth factors into tissue constructs or encapsulated ECs and tissue spheroids into biomaterials to fabricate complex 3D tissue structures with functionalized vasculature.

### 4.1. Cell-Based Approaches for Vascular Networks

The processes of vascular growth and remodeling involve ECs which line the interior of blood vessels. Currently, many studies encapsulated ECs and other supporting cell types into bioinks to fabricate vascularized tissue-engineered bone [29]. Cell-based approaches activate related cell signaling pathways by strengthening cell-cell interactions to enhance the formation of vascular networks in tissue-engineered constructs. Coculturing of different kinds of cells can achieve the goal of the prevascularization of tissue-engineered constructs [144]. Kolesky et al. fabricated cell-laden, heterogeneous, and vascularized tissue-engineered constructs by 3D Bioprinting [145]. In their study, the embedded vasculature was filled by Pluronic F127, an aqueous fugitive bioink, which could be easily printed and removed under mild conditions [146] (Figure 4a)**.** The endothelialization of embedded vasculature was conducted by the perfusion and incubation of HUVEC suspensions. Chen et al. fabricated the polydopamine-modified calcium silicate (PDACS)/poly-caprolactone (PCL) constructs with Wharton’s jelly MSCs combined with HUVEC-laden bioink [147]. The in vitro results showed that HUVECs in the bioink expressed higher levels of angiogenic proteins [147] (Figure 4b). In another study, Chiesa et al. constucted an in vitro vascularized bone model capillary-like network, using a gelatin-nanohydroxyapatite 3D bioprinted scaffold combined with MSCs and HUVECs [148].

Currently, a lot of promising techniques enhancing vascularization are utilized to combine bioprinting with vascularized bone formation. Some previous studies demonstrated that MSCs with hypoxia pre-treatment could enhance vascularization and osteogenesis in vitro an in vivo [149,150,151]. Kuss et al. [133] fabricated a polycaprolactone/hydroxyapatite (PCL/HAp) and stromal vascular fraction into tissue-engineered bone constructs, which were pre-treated in hypoxic conditions for three weeks. The results showed that short-term hypoxic conditioning could enhance the microvessel formation (Figure 4c). Non-viral gene delivery can facilitate the endogenous expression of desired therapeutic proteins, which can provide a stimulus to cells, resulting in enhanced levels of matrix production and tissue formation [152,153]. Cunniffe et al. [153] bioprinted a non-viral, and MSC-laden gene activated construct. The gene activated bioinks were fabricated by RGD-γ-irradiated alginate and nano-hydroxyapatite (nHA) complexed to plasmid DNA (pDNA) [153]. After implanted subcutaneously in vivo, gene activated MSC-laden constructs could effectively improve mineralization and vascularization. 

### 4.2. Tissue Spheroid-Based Approaches for Vascular Networks

Another approach for the formation of vascular networks involves multiple vascular cell types aggregated as multicellular vascular tissue spheroids [154]. Unlike encapsulated cells which need time to proliferate, spheroids can start with a considerably high density of cells [99]. Spheroids can also mimic the functional and architectural characteristics of target tissue. The spheroidal microcapsules permit intercellular contacts, cell aggregation, and 3D cell growth [155]. Several approaches have been applied to fabricate spheroids for bioprinting purposes, including micromolding, cell sheets, microfluidics, rotating wall vessel techniques, pellet culture, hanging drop, spinner culture, liquid overlay, and external force [99]. Tissue spheroids can fuse and assemble into macrotissues through the process of cell-to-cell adhesion. Norotte et al. [154] successfully utilized multicellular spheroids to construct tubular vascular grafts by 3D bioprinter. The results showed that the closely placed vascular tissue spheroids underwent self-assembly and tissue fusion into a branched vascular tree (Figure 5a). In another study, Tan et al. [156] have successfully utilized 3D bioprinter to robotically place tissue spheroids into the alginate mold to construct toroid-shaped vascular tissue in vitro by the fusion process of tissue spheroids consisting of smooth muscle cells and ECs. In another study, Anada et al. utilized 3D bioprinting to fabricate vascularized bone-mimetic hydrogel constructs, which consist of a central GelMA ring to mimic the bone marrow space and a peripheral GelMA ring to mimic the cortical shell, and [157]. The in vitro results demonstrated the formation of the capillary-like structures originating from the HUVEC spheroids [158] (Figure 5b).

### 4.3. Growth Factor-Based Approaches for Vascular Networks

The process of bone formation involves many influential growth factors, such as angiogenic and osteogenic factors. An alternative strategy is to fabricate the vascular network in tissue-engineered bone tissue by incorporating growth factors into 3D bioprinting constructs. There are many growth factors involved in fabricating vascularized tissue-engineered bone constructs, including vascular platelet-derived growth factor (PDGF), endothelial growth factor (VEGF), fibroblast growth factors (FGFs), epidermal growth factor (EGF), erythropoietin (EPO), transforming growth factor (TGF), hypoxia inducible factor (HIF)-1, BMP-2, and BMP-7 [144,158]. The information of growth factors used to stimulate vasculogenesis is shown in Table 3. Among these growth factors, VEGF has been identified as the most crucial signal protein to stimulate the formation of blood vessels. It is a crucial regulator of physiological vessel formation during embryogenesis [159]. Additionally, VEGF promotes both intramembranous and endochondral ossification by inducing neovascularization [160]. HIF-1 could regulate angiogenesis and vascular remodeling and plays a vital protective role in the pathophysiology of ischemic diseases [161]. EPO plays an important regulatory role in angiogenesis, especially under pathological conditions, and constitutes a crosslink between angiogenesis and hematopoiesis [162]. FGFs, PDGF, and TGF-β could stabilize newly formed blood vessels by recruiting smooth muscle cells. As bone inducers, BMPs induce transcription of numerous osteogenic genes and play a key role in the transformation of mesenchymal cells into bone [163]. Besides, as members of the BMPs, BMP-7 and BMP-2 have been approved for clinical use by the FDA [164]. In addition, stromal-derived factor-1 (SDF-1) is known to act chemotactically on ECs and thereby to enhance the process of angiogenesis [165].

Growth factors have a short half-life and are rapidly eliminated, which leads to insufficient amounts [170]. Therefore, a controlled long-term release of growth factors would be helpful in vessel formation in bone tissue constructs [171]. Currently, many growth factor delivery systems have been applied to achieve the goals of sustained release and targeted transport, such as layer-by-layer technology, hydrogel-based delivery and direct adsorption [172,173,174]. Gelatin is a natural product that is used in many kinds of FDA-approved devices. Poldervaart et al. have successfully encapsulated VEGF into gelatin and improved the degree of vascularization [175]. Compared with the single application of growth factors, recent applications also have tried to encapsulate multiple growth factors to improve the osteogenic and angiogenic ability of 3D bioprinting tissue-engineered bone [125,126]. The strongly desired characteristics of advanced bone tissue scaffolds include their ability to regulate the behavior of various cells and to mimic the structure of target tissue. Cui et al. utilized CAD to fabricate a 3D bioprinting bone construct with fluid perfused microvascular structures, and constructed a smart nanoscale release system of dual growth factors (VEGF and BMP-2) [126]. After cultured in co-cultured dynamic fluid systems, the 3D bioprinting scaffolds with dual growth factors and sequential release exhibited excellent bioactivity and vascularized bone forming potential (Figure 6a). In another study, they fabricated a biomimetic vascularized bone construct with regional immobilization of BMP-2 and VEGF [125]. The intrinsic gradient of growth factors within the engineered constructs has been proved to enhance the formation of microcapillaries [39] (Figure 6b). Byambaa et al. [30] utilized 3D bioprinting to fabricate a complex bone-like 3D architectures with vasculogenic and osteogenic niches. Moreover, to promote vascular spreading, chemically conjugated VEGF with gradient concentrations were constructed around bone niches. (Figure 6c).

### 4.4. Small Moleculers-Based Approaches for Vascular Networks

Distinct from growth factors, small molecules are low molecular weight molecules, such as lipids, metabolites, and drugs [176]. Recently, many kinds of small molecules were synthesized to enhance angiogenesis. Sildenafil, a phosphodiesterase type 5 inhibitor, not only enhances nitric oxide metabolism but has been shown to improve vascular endothelial function [177]. Lithium was also shown to have a concentration-dependent effect on early vascular development in the chick embryo area vasculosa [178]. Delivery of FTY720, a selective agonist for the sphingosine 1-phosphate receptor, has also been proven to enlarge existing arterioles and enhance the formation of new arterioles [166].

Functional peptides are much smaller than full-length growth factors, which are easier to synthesize and cheaper [179]. The functional motif KLT (KLTWQELYQLKYKGI) is a VEGF mimetic peptide and could combined to VEGF receptors and improve the migration, proliferation, and tubulogenesis of ECs [180]. Lu et al. modified hyaluronic acid hydrogels with KLT and demonstrated that the HA-KLT hydrogel could improve the spreading, and proliferation of HUVECs in vitro and promoted angiogenesis in vivo [181]. The glycine-histidine-lysine (GHK) peptide is a fragment of osteonectin, and could promote the secretion of VEGF from MSCs in alginate hydrogels [182]. In addition, Klontzas et al. demonstrated that the oxidized alginate hydrogels modified with GHK significantly improved osteogenic differentiation of encapsulated MSCs in vitro [183]. Even though the use of peptides in bioprinting is actual quite unexplored, they show great potency in the fabrication of vascularized 3D tissue-engineered bone constructs.

## 5. The Application of 3D Vascularized Models

3D bioprinting technology could precisely control complex 3D architecture and spatial distribution to fabricate 3D vascularized models, which could have great potential for the applications in drug toxicology, drug screening, and potentially disease modeling [20]. Engineered blood vessels can be integrated into 3D biomimetic tissue constructs by 3D bioprinting to mimic the drug administration process in vivo. Massa et al. utilized 3D bioprinting technology to construct a vascularized tissue-engineered model for mimicking physiological drug diffusion and drug toxicity testing [184]. The results demonstrated that the integration of vascularized tissue engineering constructs with bioreactors can helps to fabricate a new, more realistic platform that bridge the gap between in vivo and in vitro drug testing models [184]. Bhise et al. fabricated an organ-on-a-chip platform by combining a 3D bioprinted tissue-engineered construct with a bioreactor [185]. The results demonstrated organ-on-a-chip platform could be a valuable approach for drug toxicity analysis [185]. In addition, Zhou et al. utilized 3D bioprinting technology to constuct a 3D bioprinted tissue-engineered bone construct that facilitates the integration of MSCs, breast cancer (BrCa) cells, and osteoblasts [186]. BrCa cell morphology, migration, and interaction with MSCs and osteoblasts in this system were studied. The results demonstrated that 3D bioprinted tissue-engineered bone construct could be an effective disease model for investigating breast cancer bone invasion and metastasis.

## 6. Conclusions and Challenges

In recent years, 3D bioprinting has been an emerging and accurate technology to fabricate vascularized tissue-engineered bone constructs layer by layer. Combined with clinical imaging techniques, more complex biomimetic bone structures with multiple types of cells spatially arranged could be bioprinted for clinical transplantation. Although there are several reviews on 3D bioprinting, to our best knowledge, this review is the first review to examine the role of vascularization in bone tissue fabrication during the process of 3D bioprinting. In our review, we recap the current status of 3D bioprinting technology for the fabrication of vascularized tissue-engineered bone. Although successful bioprinting of various vascularized tissue-engineered bone constructs has been reported, several contemporary issues of bioprinting still require focused efforts. Firstly, there is no standard guideline for the overall process of 3D bioprinting technology. Secondly, compared with avascular cartilage tissue with a small number of chondrocytes, the structural complexity of bone tissue increases the difficulty of fabrication. Additionally, different regions of bone tissue in the human body might have different biomechanical properties and microenvironment architectures. Therefore, accurate acquisition of imaging data, computational simulation, and mathematical models might help in designing the structure of target tissue. Thirdly, unlike some tissues that contain only one type of cells, the fabrication of vascularized tissue-engineered bone needs at least two types of cells. Therefore, how to maintain cell viability throughout the process of bioprinting and ensure the exchange of nutrients and bioactive factors between host and tissue-engineered constructs are important. In our opinion, more complex and intelligent bioinks are required to sustain cell bioactivity and mimic the properties of nature tissues. Additionally, long-term cell viability, the intercellular interactions, and cell-to-material interactions after bioprinting should be monitored precisely. Fourthly, bone tissue is a hard tissue, which provides structure and support for the body. In order to fabricate biomimetic tissue-engineered bone constructs, good mechanical properties of scaffolds are necessary. In our opinion, bioinks should be modified to enhance its mechanical strength and the shape fidelity. In addition, when dealing with defects in weight-bearing areas, acellular load bearing implants followed by bioprinting cell-laden bioinks (such as [125]) might be a good solution. Finally, we believe that further developments in 3D bioprinting in the near future will undoubtedly propel the field of bone regeneration to a new height by fabricating biomimetic vascularized tissue-engineered bone constructs.

## Figures and Tables

**Figure 1 materials-13-02278-f001:**
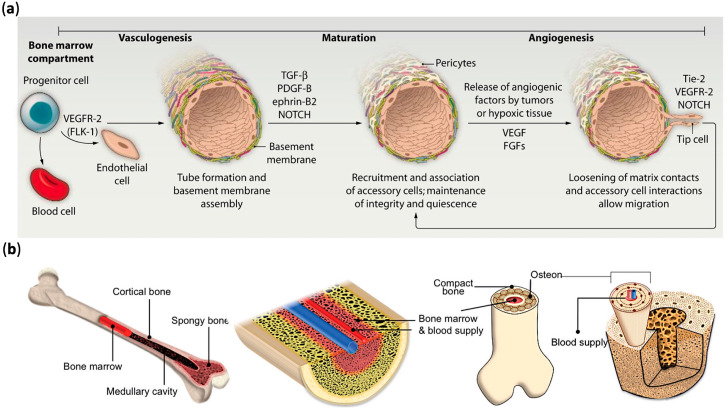
(**a**) The process of vasculogenesis and angiogenesis; (**b**) the schematic illustration of complex bone tissue with vascular structure. Reprinted with permission from References [29,30].

**Figure 2 materials-13-02278-f002:**
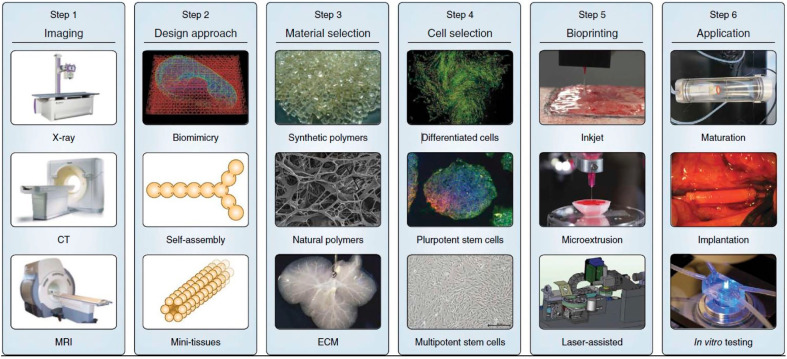
The general step-wise procedure for bioprinting 3D tissues. Reprinted with permission from Reference [12].

**Figure 3 materials-13-02278-f003:**
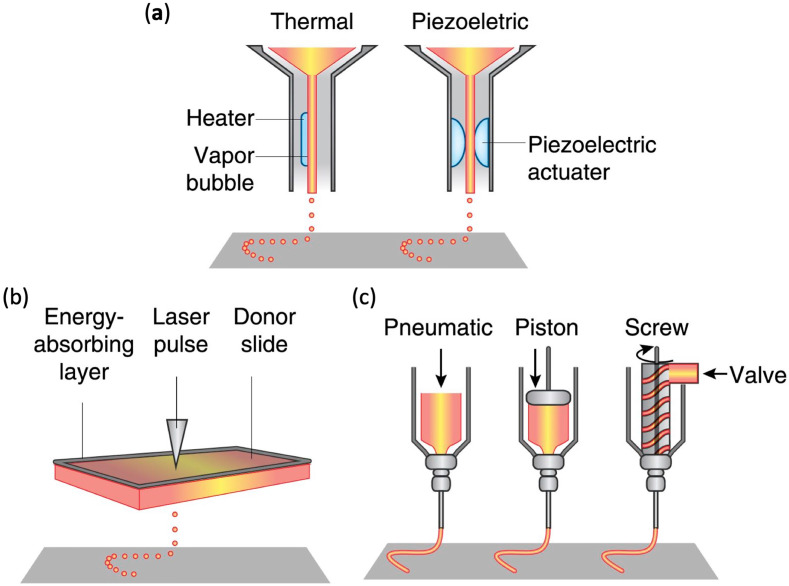
The simplified procedure of different kinds of 3D bioprinting. (**a**) Inkjet bioprinting; (**b**) laser-assisted bioprinting; (**c**) extrusion-based bioprinting. Reprinted with permission from Reference [12].

**Figure 4 materials-13-02278-f004:**
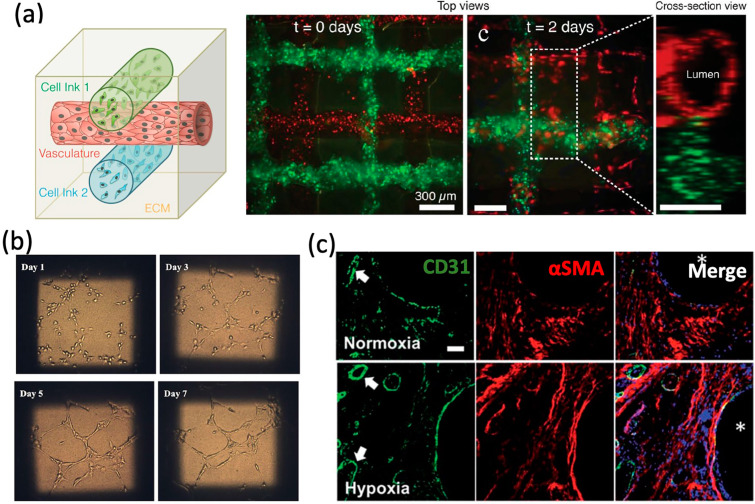
Cell-based approaches for vascular networks. (**a**) Schematic view and fluorescence images of an engineered tissue construct cultured for 0 and 2 days, respectively. (**b**) The tube formation of HUVEC-laden hydrogel/PDACS/PCL scaffold. (**c**) CD31, aSMA, and nuclei staining for 3D bioprinted constructs. Reprinted with permission from References [133,145,147].

**Figure 5 materials-13-02278-f005:**
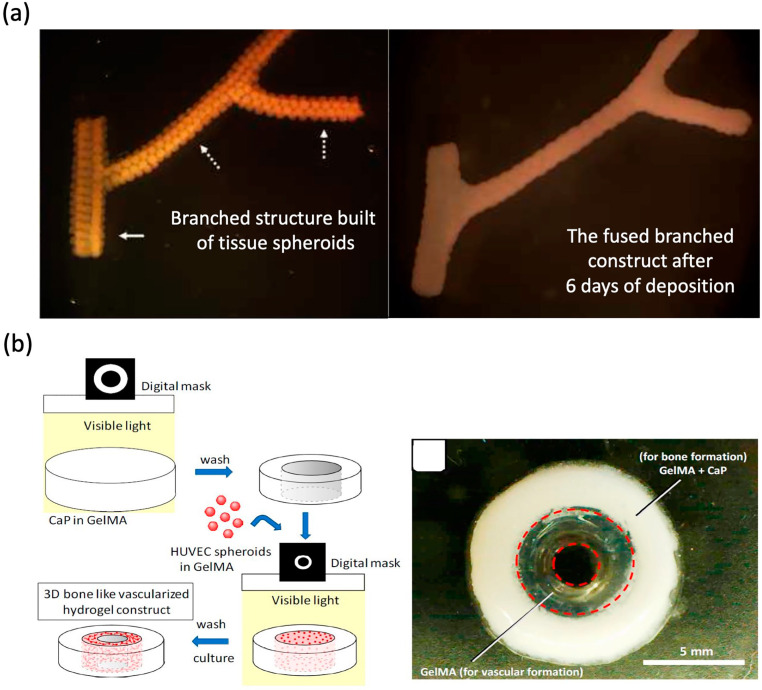
Tissue spheroid-based approaches for vascular networks. (**a**) Branched structure built of tissue spheroids and the fused branched construct. (**b**) Schematic illustration of fabrication process. Reprinted with permission from References [154,157].

**Figure 6 materials-13-02278-f006:**
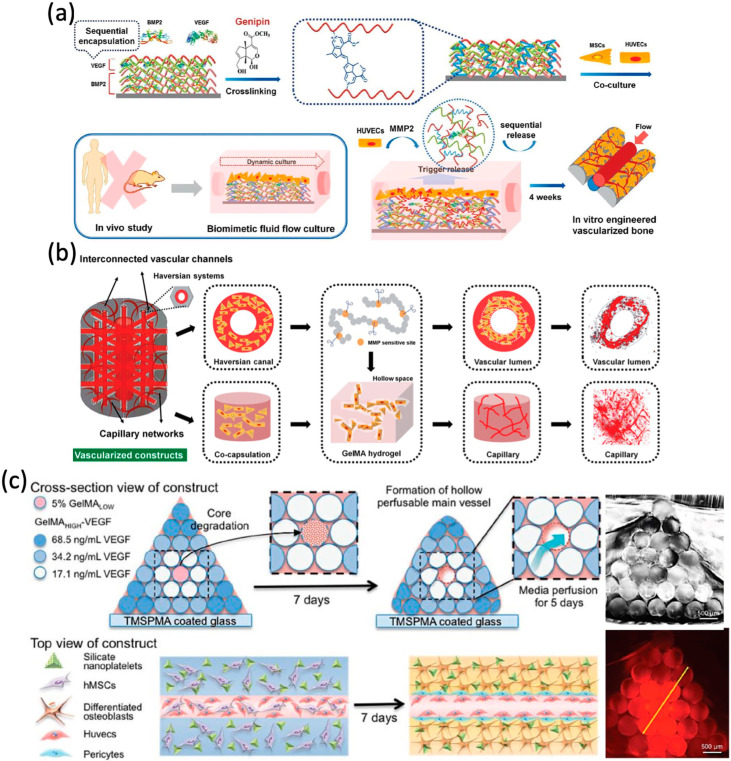
Growth factor-based approaches for vascular networks. (**a**) Schematic representation of sequential release of growth factors. (**b**) Schematic representation of microstructural design of vascularized construct. (**c**) Illustration of the bioprinting strategy for fabricating complex bone tissue architecture. Reprinted with permission from References [30,125,126].

**Table 1 materials-13-02278-t001:** Comparison of different kinds of 3D bioprinting.

Bioprinting Type	Inkjet Bioprinting	Laser-Assisted Bioprinting	Extrusion-Based Bioprinting	References
Working principle	Propels droplets of bioinks	Laser is fired to push cell from pool of bioinks	Deposition of materials by motor-driven extruder	[19,49]
Fabrication speed	Fast	Medium	Slow	[50]
Printer cost	Low	High	Medium	[51]
Cell density	Low<10^6^ cells/ml	Medium (<10^8^ cells/ml)	High, cell spheroids	[12,52]
Cell viability	>85%	>95%	40%–90%	[50,53]
Scalability	Yes	Limited	Yes	[54,55]
Resolution	High	High	Medium	[56]
Supported viscosities	3.5 to 12 mPa/s	1 to 300 mPa/s	30 to 6×10^7^ mPa/s	[57,58]
Cell type	MSCs, chondrocytes,	Fibroblasts, HUVECs, human breast cancer cells, HaCaTs, Human osteoprogenitor cells.	Chondrocytes, ASCs, MSCs, HUVECs, Neural cells, osteoblasts, Schwann cells.	[59,60,61,62]
Natural bioinks	Alginate, fibrinogen, hydroxyapatite	Alginate, collagen, matrigel	Alginate, gelatin, hyaluronic acid, agarose, chitosan, excellularized matrix	[63,64,65,66,67]
Synthetic bioinks	PCL, PEG, PVP	-	PCL, PEG, Pluronic, FG-HA	[52,68,69,70]
Target tissue	Vascular, cartilage, bone, lung	Vascular, skin, bone, adipose	Vascular, cartilage, bone, liver, brain, osteochondral tissue, cardiac tissue, nerve, aorta,	[70,71,72,73,74,75]

**Table 2 materials-13-02278-t002:** Summary of various bioinks.

Bioinks	Type	Crosslinking	Cell Type	Target Tissue	References
Silk	Natural	Enzymatic	Fibroblasts, MSCs	Bone, cartilage, brain	[111]
Chitosan	Natural	Ionic	MSCs	Cartilage	[116]
Decellularized extracellular matrix	Natural	Physical and Enzymatic	ASCs, myoblasts, hepatocytes	Liver, heart, adipose	[57,112]
Hyaluronic acid	Natural	Covalent	Osteoblasts, chondrocytes,	Bone, cartilage	[75,113]
Fibrin	Natural	Enzymatic	Chondrocytes, ECs	Vascular, cartilage	[117,118]
Collagen	Natural	Thermal	MSCs, HaCaTs, fibroblasts,	Skin, vascular, bone, cartilage, thyroid gland	[84,85]
Gelatin	Natural	Thermal, Ultraviolet	MSCs, myoblasts	Aortic valve, vascular, cartilage	[119]
Alginate	Natural	Ionic	Cartilage progenitor cells, ECs, ACSs, liver cells, MG63 cells	Vascular, liver, cartilage	[118,120,121]
Agarose	Natural	Thermal	MSCs	Cartilage	[122]
Gellan gum	Natural	Ionic	MC3T3, MSCs, Neural cells	Brain, bone	[113,123]
PEG	Synthetic	Ultraviolet	HUVECs, MSCs	Bone, vascular	[40,114,124]
PCL	Synthetic	Thermal	Chondrocytes	Cartilage	[115,121]
Pluronic acid	Synthetic	Thermal	Chondrocytes	Cartilage	[40,112]

**Table 3 materials-13-02278-t003:** Growth factors used to stimulate vasculogenesis.

Growth Factors	Gene Location (Human)	Receptor	Function	References
VEGF	Chromosome 6	Flt-1, Flk-1, KDR	Neovasculature and angiogenesis	[159]
FGF	Chromosome 8, 11, 12	FGFR1b, FGFR2b, FGFR3b, FGFR4	Embryonic development and angiogenesis	[166]
PDGF	Chromosome 22	PDGFRα and β	Maturation of vasculature	[167]
TGF	Chromosome 19	TGF receptor	Vascular invasion	[166]
Angiopoietin-1	Chromosome 8	Tie-2 receptor	Enhance vasculature stability	[168]
BMP	Chromosome 12	BMPR1A, BMPR1B	Regulate angiogenesis and VEGF secretion	[169]

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
