# Peer review of "3D Bioprinting for Vascularized Tissue-Engineered Bone Fabrication"

_materials, 2020, doi:10.3390/ma13102278_

Round 1

Reviewer 1 Report

There are several issues in the manuscript that should be addressed before further consideration for publication. 1. There has been quite a few review on 3D bioprinting of bone. What is the new contribution from this review? - Lim et al. (2019), Three-dimensional bioprinting for bone and cartilage transplantation, Annals of Joint 4 - Ibrahim (2018), 3D bioprinting bone in 3D Bioprinting for Reconstructive Surgery Techniques and Applications 2. Different 3D bioprinting techniques have been reviewed and summarised by a few publication. What is the new contribution from this work? 3. For vacularisation, any consideration of mulit-materials printing using these techniques? What are some of the limitations and challenges? - Lee et al. (2020), Bioprinting of Multimaterials with Computer-aided Design/Computer -aided Manufacturing, International Journal of Bioprinting 6 (1), 245 4. Any consideration of direct 3D bioprinting compared to tissue engineering (scaffolding using 3D printing)? 5. Some of the figures are not clear. Please obtain images of higher resolutions

Reviewer 2 Report

This is generally a thorough and very well written review on 3D bioprinting for vascularised tissue-engineered bone formation and it is of significant value to the scientific and clinical community. There are  however 2 areas that I feel could be more significantly addressed within the manuscript

  1. a more thorough discussion (a sub-heading) specifically on the influence of printing and encapsulation on cell viability
  2. some discussion on the limitations of mechanics - are load bearing implants viable?

Reviewer 3 Report

Dear Authors,

After the review process, I have several comments: you should improve de figures' quality and declare if the figures are an authors contribution or they are copied from other sources; you should detail the possibility to use bioprinted 3D vascularized model for drug toxicity analysis.

Best regards!

Reviewer 4 Report

The review titled '3D bioprinting for...' by U. Ritz et al. compiles methods of tissue engineering. The manuscript has 23 pages and 159 references. It can be useful for some scientist to find in a single document relevant information. However, some issues should be addressed:

-quality of pictures must be improved;

-a blank should be between word and reference;

-references 70 and 139 are incomplete;

-some references lack DOI numbers;

-some references use hyperlinks for DOI.

Therefore, the work can be accepted for publication after these correcctions are made and the manuscript is improved.

Reviewer 5 Report

Figure 1, 2, 6 require higher resolution as the words are difficult to read. 

Table 1: supported viscosity value for extrusion-based printing seems to have superscript error? 

The author raised the issue of poor directionality of vessel formation. However, synthetic body parts with complex shape and structure (e.g. ear) can be successfully produced. I suggest the author to comment on comparing the 3D printing of ears vs bones etc, which will be beneficial for future research. 

Overall, the review is informative and reasonably up to date. The manuscript is easy to read. I suggest publication with some additional information stated above. 

Reviewer 6 Report

This descriptive review report the application of 3D bioprinting technology in the vascularized tissue-engineered bone fabrication; the mechanisms of vascular formation, the process of 3D bioprinting, and the current development of bioink properties are also been reported.

INTRODUCTION

The intrinsic limitations of traditional tissue engineering technologies to recreate complexes and similarities from native tissues must be well reported, in order to better comprehend their restricted applications.

The differences between 3D bioprinting technology and conventional scaffold-based approaches must be well reported, and the advantages offered by this new technology must to be highlighted.

All the figures (1,2,3,4,5,6)  has been reprinted with the permission of the author; even if the permission to use previous published images has been acquired, original images would be preferred.

CONCLUSIONS

There are not sufficient evidence about the standards in manufacturing processes necessary for clinical translation and to decrease the product-development time. The conclusions are too generic and  based only on the opinion of the authors; a stronger scientifc support must be given in order to have more relliable informations about this new technology.

The challenges for a future application of this new technique are not well reported and must be highlighted.

Reviewer 7 Report

I would like to thank the authors for submitting this interesting review paper. In general the review is comprehensive and i have only minor comments that would further enhance the content of the manuscript:

1) I would like to see a more comprehensive section on growth factors and small molecules used to stimulate vasculo- and angio-genesis. Please increase the section. Please check (and cite) review papers which have summarised drugs used to enhance bone healing and vasculogenesis in bone and include all drugs and growth factors not mentioned (such as EPO, sildenafil etc.). Such review papers are: 

  • Bhise NS et al. Expert Opin Drug Deliv, 2011, 8(4): 485-504

2) please also add a table on the growth factors used to stimulate vasculogenesis

3) line 245: please correct "hyaluronic aci"

4) I would like to see a paragraph on peptides used to stimulate vasculogenesis on biomaterials (citing at least the following relevant papers) such as:

  • KLT (e.g. Lu J, Regenerative Biomaterials, Volume 6, Issue 6, December 2019, Pages 325–334)
  • GHK (Klontzas et al. Acta Biomaterialia, 2019, 88: 224-240)

Reviewer 8 Report

This is an interesting review of the literature on 3d Bioprinting methods about bbe fabrication

Some criticisms are present:

-Line 27 replace will with were

-In the Introduction, too short in my opinion, general considerations on the use of nanotechnologies in the medical field should be added. I recommend, for the sector of my relevance, to include the following scientific work in the references and in the discussion:

Chieruzzi M, Pagano S, Moretti S, Pinna R, Milia E, Torre L, Eramo S.

Nanomaterials for Tissue Engineering In Dentistry. Nanomaterials (Basel). 2016

Jul 21;6(7). pii: E134.

-Line 59 always use verbs in the past tense

-Figure 1a The figure is not clear and does not help understanding the text

-Figure 2 is totally blurry to increase the resolution

. -Line 175 the authors report a single study using the Laser-assisted bioprinting technique. Absolutely insufficient

-More generally the work in question is very interesting and well built. However, there is no evidence of how the review work was performed, with which search engines, with which inclusion and exclusion criteria, etc. There is also no assessment of the Quality assessment of the studies included in the review

It therefore seems more of an excellent chapter of text rather than a scientific review.

Starting from these considerations, I believe that it can be accepted but only after a profound major revision of the above criteria

Round 2

Reviewer 1 Report

NA

Reviewer 6 Report

The revised version of the manuscript has been developed overcoming the limits of the original version of the paper.

The only limit of the revised version is related to the figures,  that are almost all derived from other studies.

Reviewer 8 Report

all questiones were added

i reccomend work acceptation